# Relationship between Eye Frailty and Physical, Social, and Psychological/Cognitive Weaknesses among Community-Dwelling Older Adults in Japan

**DOI:** 10.3390/ijerph192013011

**Published:** 2022-10-11

**Authors:** Masafumi Itokazu, Masahiro Ishizaka, Yoshikazu Uchikawa, Yoshiaki Takahashi, Takahiro Niida, Tamaki Hirose, Akihiro Ito, Akihiro Yakabi, Yoshiaki Endo, Yohei Sawaya, Tatsuya Igawa, Kaoru Kobayashi, Tsuyoshi Hara, Miyoko Watanabe, Akira Kubo, Tomohiko Urano

**Affiliations:** 1Department of Physical Therapy, School of Health Sciences, International University of Health and Welfare, Otawara 324-8501, Tochigi, Japan; 2Department of Orthoptics and Visual Sciences, School of Health Sciences, International University of Health and Welfare, Otawara 324-8501, Tochigi, Japan; 3Department of Ophthalmology, Dokkyo Medical University Nikko Medical Center, Nikko 321-2593, Tochigi, Japan; 4Department of Geriatric Medicine, School of Medicine, International University of Health and Welfare, Narita 286-8686, Chiba, Japan

**Keywords:** elderly, eye frailty, psychological/cognitive weaknesses

## Abstract

This study investigated the relationship between eye frailty and physical, social, and psychological/cognitive weaknesses among older adults in Japan. The participants were 192 community-dwelling older adult women. We measured handgrip strength, walking speed, and skeletal muscle mass; additionally, their physical, social, and psychological/cognitive frailties were surveyed using questionnaires. Eye frailty self-checks were used to assess eye frailty. Exploratory and confirmatory factor analyses were employed to verify the validity of the eye frailty self-checks. Eye frailty prevalence and related factors were investigated by conducting a binomial logistic regression analysis, with eye frailty as the dependent variable. The factor analysis results showed that a model could be constructed with the two factors of visual acuity or contrast and visual field. The model’s goodness of fit was acceptable, supporting the validity of the self-checking construct. The Kihon checklist was the only variable with a significant relationship to eye frailty. Regarding the relationship between eye frailty and subordinate items of the Kihon checklist, social withdrawal [odds ratio (OR) 2.437, 95% confidence interval 1.145–5.188], cognitive function (OR 2.047, 95%CI 1.051–3.984), and depressed mood (OR 1.820, 95%CI 1.163–2.848) were significant. Eye frailty can be considered a factor reflecting the existence of social and psychological/cognitive frailties.

## 1. Introduction

The concept of “frailty” is important for early detection and response to conditions that inhibit the healthy life expectancy of older adults. There are different ways of thinking about frailty, such as the phenotype model [1] and the deficit accumulation model [2]. Frailty can be caused not only by physical problems, such as sarcopenia, osteoarthritis, and osteoporosis, but also by psychological problems, such as cognitive dysfunction and depression, and social problems, such as living alone and economic deprivation [3]. In addition, Gobbens et al. [4] “Integral conceptual model of frailty” shows that physical, psychological, and social frailty interact with each other to lead to negative health outcomes. In other words, it is important to comprehensively examine frailty from physical, social, and psychological/cognitive perspectives to understand the factors that affect an individual’s health.

In recent years, the relationship between age-related decline in sensory function and frailty has been the focus of much attention [5]. Among them, most of the studies reporting the relationship between visual function and frailty suggest that the decline in visual function due to aging or ocular disease is one of the important factors in the development of frailty [6,7,8,9,10]. Previous studies have also reported a relationship between visual acuity and skeletal muscle mass in older adults [11], a higher probability of frailty in older adults with uncorrected refractive error [12], and a relationship between a history of eye disease and falls [13,14]. On the other hand, the presence of visual impairment has also been shown to be associated with social activity and social isolation [14,15,16]. A study [17] that divided the components of visual impairment into visual acuity and contrast sensitivity and examined the prevalence of frailty according to the presence or absence of each impairment found a relationship between contrast sensitivity ability and the prevalence of frailty. Thus, age-related changes in visual function have a significant impact on the lives of older adults and are an important predictor of frailty, and a major inhibitor of healthy life expectancy.

Eye frailty is a conceptualization of the changes in visual function with aging. According to the Japan Ophthalmology Awareness Council [18], it is a “condition wherein visual function decreases due to the combination of various external and internal factors with increased eye fragility due to aging, or a state wherein the risk of such an occurrence is high.” The eye frailty self-check has been proposed to enable individuals to check their eye frailty as a screening test. These consist of ten self-reported questions, each of which corresponds to the major prodromal and early symptoms of age-related diseases (cataracts, age-related macular degeneration, dry eye, etc.). Decreased visual function is an obstacle to the conduction of independent activities of daily living among older adults; however, eye frailty is a relatively new concept, and the validity of screening results has not been verified. The relationship between eye frailty and physical, social, and psychological/cognitive frailties remains unclear. Therefore, the purpose of this study was to confirm the construct validity of the eye frailty self-check and to clarify the relationship between eye frailty and physical, social, psychological, and cognitive weaknesses.

## 2. Materials and Methods

### 2.1. Participants

The participants for this study were recruited from those who voluntarily participated in a care prevention project publicly implemented by the city of Otawara, Tochigi Prefecture, from June 2021 to January 2022. The project was recruited through a city newsletter, and 225 older adults (28 males and 197 females; mean age: 79.0 ± 7.4 years) voluntarily participated in this study. Men and participants with missing data were excluded, and a total of 192 women (mean age: 79.2 ± 7.2 years) were included in the analysis.

### 2.2. Survey Items

We collected age information and measured height and weight as basic information. Regarding physical function, body composition was measured using handgrip strength, usual walking speed, and the bio-electrical impedance analysis (BIA) method. For handgrip strength, the handgrip strength values were measured using a handgrip strength meter (D-TKK5401, Takei Scientific Instruments Co., Ltd., Niigata, Japan), and the maximum value was considered. For the usual walking speed, the typical time to cover a 4 m walking path was measured twice using a stopwatch, and the fastest value was considered. For body composition measurement, the limb skeletal muscle mass was measured using a multi-frequency body composition meter (MC-780A, Tanita Co., Ltd., Tokyo, Japan), and divided by the square of the height to obtain the skeletal muscle mass index (SMI), which was used as the input value.

Participants’ frailty status was comprehensively assessed using physical frailty, social frailty, and the Kihon checklist. Physical frailty was defined by Fried’s phenotype, and the Revised Japanese Cardiovascular Health Study (Revised J-CHS) was used, and the criteria were as follows: those who had one or two of the following five items: weight loss, muscle weakness, fatigue, walking speed, and physical activity were classified as physical pre-frail, and those with three or more items were classified as physical frailty [19]. Social frailty was assessed using the criteria of Makizako et al. [20]. Five questions were given to determine this: going out less frequently than in the last year (yes), visiting friends sometimes (no), feeling like helping friends or family (no), living alone (yes), and talking with someone every day (no). Individuals who fulfil one of these five questions were social pre-frailty, two or more were classified as social frailty. The Kihon checklist consists of 25 questions about older adults’ living conditions and physical functioning, social status, and psychological/cognitive weaknesses, and is a comprehensive measure of physical frailty, social frailty, and psychological/cognitive weaknesses [8].

The questionnaire proposed by the Japanese Council for Eye Health Care Awareness was used to assess eye frailty, and participants who answered “Yes” to two or more of the 10 questions were considered to have eye frailty [18].

### 2.3. Analyses

We verified the validity of the 10-item eye frailty self-check construct. First, the correlation matrix between the items of the eye frailty self-check was obtained using Spearman’s rank correlation analysis. Subsequently, we conducted exploratory factor analysis using the maximum likelihood method and Promax rotation to construct the model and determine the number of factors and model. In the model, the superordinate concept was eye frailty, and the obtained factors were independent subordinate concepts. The exclusion of items in the model configuration was also avoided to the extent possible, considering the independence and importance of the individual questions. Therefore, items with a factor loading of 0.2 or more were retained. The number of factors was determined based on the Kaiser Guttman criterion—that is, factors with eigenvalues of 1.0 or higher were retained. Next, we constructed a model using the factors derived from the exploratory factor analysis and attempted a confirmatory factor analysis using structural equation modeling. Indicators for the model’s goodness of fit involved calculation of the comparative fit index (CFI), and root mean square error of approximation (RMSEA). A CFI of 0.9 or more [21,22] and RMSEA of 0.06 or less [21] were used as guidelines for model acceptance.

This study calculated the prevalence of eye frailty among the older adults surveyed. Participants were divided into two groups according to eye frailty. Furthermore, an unpaired t-test was used to clarify the differences in physical function between the groups, and compare participants’ age, height, weight, body mass index (BMI), Skeletal Muscle mass Index (SMI), calf circumference, handgrip strength, and usual walking speed. The differences in the prevalence of physical and social frailty and the number of people who experienced falls according to the presence or absence of social frailty were verified using a χ2 test.

To investigate the relationship between the presence or absence of eye frailty, physical function factors, and physical and social frailties, we set the two groups of “eye frailty” and “healthy” as the dependent variables; for independent variables, we set age, height, BMI, SMI, calf circumference, handgrip strength, usual walking speed, Kihon checklist, revised J-CHS, social frailty criteria by Makizako et al. [20], and the presence or absence of falls in the previous one-year period. We then conducted binomial logistic regression analysis.

The Kihon checklist includes 25 questions concerning the living conditions of older adults and their physical function, social status, and psychological/cognitive weakness [23]. It covers the seven domains of “IADL (Q1–5)”, “motor function (Q6–10)”, “undernutrition (Q11, 12)”, “oral function (Q13–15)”, “social withdrawal (Q16, 17)” “cognitive function (18–20),” and “depressive mood (21–25)” [23]. To clarify the relationship between the presence or absence of eye frailty and the subordinate items of the Kihon checklist, the former was set as the dependent variable and the latter as the independent variable, and a binary logistic regression analysis was conducted. IBM SPSS statistics 27 and Amos 24.0 were used for all statistical analyses, with the significance level set to 5%.

## 3. Results

### 3.1. Exploratory and Confirmatory Factor Analysis of Eye Frailty Self-Check Test

The results of Spearman’s rank correlation analysis are shown in Table 1. No combinations showed particularly strong correlations among the items. The exploratory factor analysis was conducted based on the Kaiser Guttman criterion, and it was considered adequate until the second factor, which had an eigenvalue of 1.0 or higher. The Kaiser-Meyer-Olkin sample validity measurement was 0.825, the *p*-value of Bartlett’s test of sphericity was <0.001, and the validity of factor analysis was confirmed. When estimating commonality, it was confirmed there were no items with commonality exceeding one.

Analysis results showed the following: for the first factor, items 1, 2, 3, 5, 6, 7, and 9 had high factor loadings and expressed the factor representing “visual acuity or contrast sensitivity.” For the second factor, items 4, 8, and 10 had high factor loadings and expressed the factor representing the “visual field.” A moderate correlation (r = 0.609) was found between the first and second factors (Table 2).

The model was constructed with eye frailty as the superordinate concept and the two factors obtained by exploratory factor analysis as the subordinate concept. Structural equation model results showed CFI = 0.973, AGFI = 0.936, RMSEA = 0.037 and SRMR = 0.042 (Figure 1). The path coefficients of the correlation between eye frailty and each factor ranged from 0.42–0.98.

### 3.2. Comparison of Prevalence of Eye Frailty and Physical Function

Of the 192 participants, 143 (74.5%) were classified as having eye frailty (Table 3). Comparisons with the group of 49 participants judged as having no eye frailty regarding age, height, weight, BMI, SMI, calf circumference, handgrip strength, and usual walking speed showed that only walking speed (*p* = 0.02) exhibited a significant decrease in the eye frailty group.

### 3.3. Investigation of Factors Related to Eye Frailty

Binomial logistic regression analysis was conducted to analyze the presence or absence of eye frailty, physical function factors, and the strength of the relationship with physical and social frailty (Table 4). The Kihon checklist was selected as the variable affecting the presence or absence of eye frailty (*p* < 0.001 in model χ2 test). The OR of the Kihon checklist was 1.385 [95% confidence interval (CI): 1.160–1.654]. The Hosmer-Lemeshow test results of this model showed the goodness of fit at *p* = 0.951, with a discriminative predictive value that lay between the predicted value and measured value of 76.6%.

### 3.4. Investigation of the Relationship between Eye Frailty and Subordinate Items of the Kihon Checklist

We conducted a further binomial logistic regression analysis to verify the relationship between the presence or absence of eye frailty and the subordinate seven items of the Kihon checklist (Table 5). Social withdrawal, cognitive function, and depressed mood were selected as variables that affected the presence or absence of eye frailty (*p* < 0.001 in model χ2 test). The OR was 2.437 (95%CI: 1.145–5.188) for social withdrawal, 2.047 (95%CI: 1.051–3.984) for cognitive function, and 1.820 (95%CI: 1.163–2.848) for depressive mood. The Hosmer-Lemeshow test results of this model showed the goodness of fit at *p* = 0.907, with a discriminative predictive value that lay between the predicted value and measured value of 77.1%.

## 4. Discussion

This study conducted assessed the visual function of community-dwelling older adults using the eye frailty self-check. The percentage of those judged to have eye frailty based on positive responses to two or more items among the total 10 items was 74.5%; thus, approximately three-quarters of the sample had some concerns with their visual function in daily life situations. There are no previous reports on the prevalence of eye frailty; however, a systematic review by Bourne et al. [24] in 2017 suggests that approximately 60% of older adults over the age of 70 had visual impairment. Nevertheless, the rate of eye examinations among older adults has been reported to be low [25], suggesting that many older adults are living with age-related visual impairment that is overlooked.

Although eye frailty is a relatively new concept, it has long been known that age-related declines in visual acuity and contrast sensitivity are related to frailty, and the diseases that cause them include cataract, glaucoma, and age-related macular degeneration [17,24,26]. At present, evaluating these items requires specialists to perform visual acuity and contrast sensitivity tests using special equipment, and there is no self-administered test battery to simplify the screening process. This study conducted a factor analysis to verify the validity of the eye frailty self-check construct and classified the 10 questions into two factors. We created a model wherein the factor of visual acuity or contrast sensitivity included seven of the 10 questions (1, 2, 3, 5, 6, 7, and 9), and the factor of the visual field comprised three questions (4, 8, and 10); the model’s goodness of fit was acceptable, with CFI = 0.973, AGFI = 0.936, RMSEA = 0.037, SRMR = 0.042, and the factors were considered valid. The exploratory and confirmatory factor analysis results confirmed the validity of the eye frailty self-check construct. Hence, the Eye Frailty Self-Check can be easily introduced into regular health checkup programs for community-dwelling older adults and is useful for screening visual function.

We also analyzed the relationship between the presence or absence of eye frailty and participants’ physique, physical and social frailty, Kihon checklist, and fall history. Results showed that only the Kihon checklist had a significant relationship. The OR was 1.385, and the 95%CI was 1.160–1.654; a one-unit increase in a Kihon checklist item resulted in the probability of eye frailty increasing by 1.385. The Kihon checklist has seven domains as subordinate items, and the three domains [4] of physical, social, and psychological/cognitive frailties can be comprehensively evaluated based on the checklist [23].

Next, we conducted a binomial logistic regression analysis of the relationship between the presence or absence of eye frailty and the seven subordinate domains of the Kihon checklist. Results showed that social withdrawal (OR: 2.437, 95%CI: 1.145–5.188), cognitive function (OR: 2.047, 95%CI: 1.051–3.984), and depressed mood (OR: 1.820, 95%CI: 1.163–2.848) were significant related factors. Comparison of the OR showed that social withdrawal, reflecting social frailty, was the most significantly related, followed by cognitive function and depressive mood, which correspond to psychological and cognitive frailty, respectively.

Previous research has reported that social withdrawal reflects social frailty, and that cognitive function and depressive mood are items corresponding to psychological or social frailty. Fujiwara et al. [27] suggested that older adults’ tendency to withdraw from social activities is linked to a decline in their living function; decreased visual function among older adults reduced their participation in community activities [28] and reduced the frequency of meeting friends [29], which suggests a relationship between visual function and social frailty. Furthermore, research on the relationship between psychological/cognitive weakness and visual impairment has shown that older adults with visual impairments showed significantly lower Mini-Mental State Examination values than those without impairments [30], as well as a high tendency for depression [19,20]. On the other hand, Nelson et al. [31] and Hikichi et al. [32] reported that older adults who participate in social interactions have a higher cognitive and physical function, are more active in social activities such as volunteering and meeting neighbors and family, and are less likely to require long-term care in the future. These findings indicate a strong relationship between visual function and social activity among older adults, and the results of this study indicate that visual impairment among older adults is significantly associated not only with physical frailty, as reported in previous studies [11,12], but also with social frailty and psychological and cognitive weakness.

This study’s results verified the validity of the eye frailty self-check construct as a screening method for evaluating the decrease in visual function with aging, finding that it has acceptable validity. Furthermore, eye frailty showed a significant relationship with the Kihon checklist and a relationship with the subordinate items of social withdrawal, cognitive function, and depressed mood. Thus, eye frailty can be considered a factor that reflects the presence of social frailty and psychological and cognitive frailty among community-dwelling older adults. Gobbens et al. proposed the integral conceptual model of frailty [4] and showed that the interaction of physical, psychological, and social frailty leads to negative health outcomes. To prevent this, the decline in age-related physical and mental functions and social activities should be anticipated at an early stage and necessary support and prevention programs should be provided. Our study demonstrates that screening for visual function may help anticipate the future onset of frailty and contribute to extending the healthy life expectancy of community-dwelling older adults.

There are several limitations to this study. First, participant recruitment was conducted as part of a city-sponsored health check-up program, so information on participants’ educational and social backgrounds was not available. Second, because of the large gap in the number of men and women participants in the health screening program, only women participants were included in this study for analysis. The results may differ between men and women, who have many differences in physiology, lifestyle, and social activities. Furthermore, the results of the Eye Frailty Self-Check were based on self-report, and no expert assessment of visual acuity, contrast sensitivity, or visual field was available. Additionally, there was no information on diseases that lead to visual impairment, such as glaucoma, diabetic retinopathy, or age-related macular degeneration, making it impossible to distinguish which ophthalmologic diseases are responsible. Finally, it should be recognized that many issues need to be addressed for clinical application, such as the identification accuracy of the eye frailty checklist and the calculation of cutoff values. However, we believe that we were able to demonstrate an association with confinement, cognitive function, depressive mood, and the possibility of screening for risk leading to these conditions.

## 5. Conclusions

The validity of the eye frailty self-check was demonstrated, and the prevalence of eye frailty among community-dwelling older adults was 74.5%, with a high association with social withdrawal, cognitive function, and depressive mood in the Kihon check list. We believe that this preliminary study provides a basis for further research on the characteristics of older adults with eye frailty and its relationship to physical and social frailty and cognitive-mental aspects.

## Figures and Tables

**Figure 1 ijerph-19-13011-f001:**
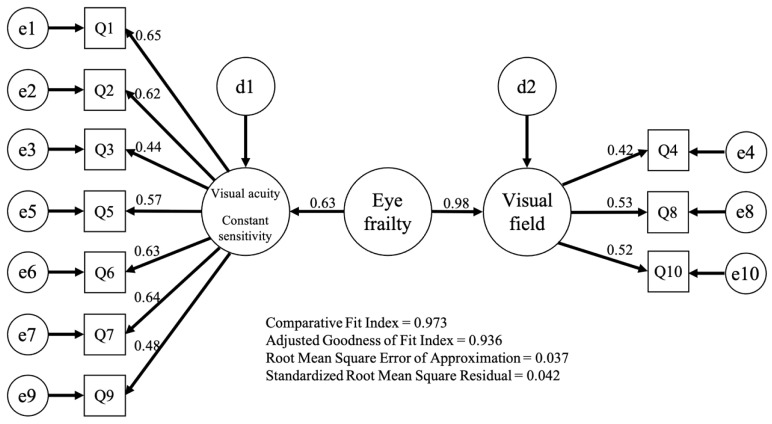
Confirmatory factor analysis in eye frailty self-check.

**Table 1 ijerph-19-13011-t001:** Correlation matrix among eye frailty self-check items.

	Q1	Q2	Q3	Q4	Q5	Q6	Q7	Q8	Q9	Q10
Q1. Eyes become tired more easily	—	0.458 **	0.347 **	0.109	0.363 **	0.395 **	0.412 **	0.139	0.325 **	0.139
Q2. It can sometimes be challenging to see in the evening		—	0.177 *	0.132	0.376 **	0.435 **	0.379 **	0.173 *	0.262 **	0.173 *
Q3. Opportunities to read newspapers or books for extended periods are now less often			—	0.176 *	0.318 **	0.242 **	0.309 **	0.108	0.161 *	0.177 *
Q4. Sometimes, the table gets dirty when eating				—	0.168 *	0.127	0.206 **	0.221 **	0.205 **	0.221 **
Q5. Often, I feel I cannot see well even with glasses					—	0.277 **	0.328 **	0.327 **	0.318 **	0.216 **
Q6. Often, I feel it is too bright						—	0.435 **	0.215 **	0.369 **	0.215 **
Q7. I cannot see clearly sometimes without blink							—	0.234 **	0.264 **	0.270 **
Q8. Straight lines appear wavy at time								—	0.143 *	0.163 *
Q9. I have sometimes felt stairs were dangerous									—	0.108
Q10. I have overlooked traffic lights or road signs										—

* *p* < 0.05, ** *p* < 0.01.

**Table 2 ijerph-19-13011-t002:** Exploratory factor analysis in eye frailty self-check.

	Factor Loading	
	Factor 1	Factor 2
	Visual Acuity, Contrast Sensitivity	Visual Field
Q1. Eyes become tired more easily	0.813	−0.185
Q2. It can sometimes be challenging to see in the evening	0.678	−0.066
Q6. Often, I feel it is too bright	0.606	0.034
Q7. I cannot see clearly sometimes without blink	0.504	0.189
Q9. I have sometimes felt stairs were dangerous	0.413	0.097
Q5. Often, I feel I cannot see well even with glasses	0.380	0.278
Q3. Opportunities to read newspapers or books for extended periods are now less often	0.341	0.143
Q4. Sometimes, the table gets dirty when eating	−0.075	0.507
Q8. Straight lines appear wavy at time	0.004	0.481
Q10. I have overlooked traffic lights or road signs	0.048	0.399

**Table 3 ijerph-19-13011-t003:** Demographic data according to healthy and eye frailty groups.

		Robust	Eye Frailty	*p* Value
Total, n (%)		49 (25.5)	143 (74.5)	
Age		79.0 ± 5.9	79.8 ± 6.7	0.44
Height (cm)		147.7 ± 5.5	148.6 ± 6.1	0.33
BMI (kg/m^2^)		23.8 ± 3.3	23.7 ± 3.1	0.82
SMI (kg/m^2^)		6.41 ± 0.64	6.37 ± 0.67	0.76
Calf circumference (cm)		33.7 ± 3.0	33.3 ± 2.9	0.48
Handgrip strength (kg)		23.1 ± 4.1	21.9 ± 4.2	0.08
Usual walking speed (m/s)		1.30 ± 0.22	1.20 ± 0.34	0.02 *
	Robust	30 (61.2)	44 (30.8)	
Physical frailty, N (%)	Pre-frailty	18 (36.7)	79 (55.2)	
	Frailty	1 (2.0)	20 (14.0)	0.0001 ^†^
	Robust	22 (44.9)	38 (26.6)	
Social frailty, N (%)	Pre-frailty	23 (46.9)	83 (58.0)	
	Frailty	4 (8.2)	22 (15.4)	0.0001 ^†^
fall, N (%)	No fall	38 (77.6)	109 (76.2)	
	Fall	11 (22.4)	34 (23.8)	0.85

* Unpaired t-test, ^†^ χ2 test. Mean ± standard deviation. BMI, body mass index. SMI, skeletal muscle mass index.

**Table 4 ijerph-19-13011-t004:** Variables affecting the presence or absence of eye frailty.

	B	SE	Wald	*p* Value	Odds Ratio	95%CI	
						Lower	Upper
Age	−0.062	0.038	2.642	0.104	0.940	0.872	1.013
Height	0.088	0.048	3.424	0.064	1.092	0.995	1.199
BMI	0.028	0.111	0.065	0.798	1.029	0.828	1.278
SMI	0.094	0.367	0.066	0.797	1.099	0.536	2.256
Calf circumference	−0.090	0.130	0.479	0.489	0.914	0.709	1.179
Handgrip strength	−0.083	0.066	1.552	0.213	0.921	0.808	1.049
Usual walking speed	−0.182	0.909	0.040	0.842	0.834	0.140	4.953
Kihon check list	0.326	0.090	13.013	0.000 *	1.385	1.160	1.654
Physical frailty (J-CHS)	0.216	0.335	0.416	0.519	1.242	0.643	2.396
Social frailty	0.225	0.237	0.903	0.342	1.253	0.787	1.994
Fall	−0.269	0.450	0.356	0.551	0.764	0.316	1.847
(Constant)	−5.238	7.872	0.443	0.506	0.005		

* *p* < 0.05. SE, standard error; CI, confidence interval; BMI, body mass index; SMI, skeletal muscle mass index; J-CHS, revised Japanese version of the Cardiovascular Health Study criteria; variable selection procedure, simultaneous. Model χ2 test < 0.001; Hosmer-Lemeshow test = 0.951; Percentage of correct classifications = 76.6%.

**Table 5 ijerph-19-13011-t005:** Kihon checklist subordinate items affecting the presence or absence of eye frailty.

	B	SE	Wald	*p* Value	Odds Ratio	95%CI	
						Lower	Upper
IADL	−0.471	0.324	2.112	0.146	0.624	0.331	1.178
Physical functions	0.265	0.186	2.038	0.153	1.304	0.906	1.878
Nutritional status	−0.045	0.679	0.004	0.947	0.956	0.253	3.617
Oral function	0.122	0.256	0.227	0.633	1.130	0.685	1.864
Social withdrawal	0.891	0.385	5.343	0.021 *	2.437	1.145	5.188
Cognitive function	0.716	0.340	4.440	0.035 *	2.047	1.051	3.984
Depression mood	0.599	0.228	6.877	0.009 *	1.820	1.163	2.848
(Constant)	0.039	0.456	0.007	0.932	1.039		

* *p* < 0.05; SE, standard error. CI, confidence interval. Variable selection procedure, simultaneous. Model χ2 test < 0.001. Hosmer-Lemeshow test = 0.907. Percentage of correct classifications = 77.1%.

## Data Availability

There are no linked research datasets for this study. The authors do not have permission to share the data.

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
