# Peer review of "Relationship between Eye Frailty and Physical, Social, and Psychological/Cognitive Weaknesses among Community-Dwelling Older Adults in Japan"

_ijerph, 2022, doi:10.3390/ijerph192013011_

Round 1

Author Response

Dear Reviewer 1

The authors would like to thank the reviewers and editors for their time and effort to review this manuscript.  Please consider the attached manuscript that has been revised according to the reviews.  Response to each reviewer's comments follows the attached.

Reviewer 2 Report

I congratulate the authors for an interesting study and a new approach of frailty. Eye frailty is a less known concept and the tools for its diagnose less known. 

The introduction is extensive, with relevant information.

In materials and methods section

- the new proposed tool (questionnaire)  - i cannot find the explanation for determining frailty; on which score / questions we consider a person with eye frailty? Please elaborate

In results section, the data is extensively described, with relevant tables.

In discussion section please accentuate the fact that the results are only on female population. It can only be speculate on men. There are some normal physiological differences between men and women and the results may differ.

The bibliography may benefit from some more updated data.

Author Response

Dear Reviewer 2

The authors would like to thank the reviewers and editors for their time and effort to review this manuscript.  Please consider the attached manuscript that has been revised according to the reviews.  Response to each reviewer's comments follows the attached.

Round 2

Reviewer 1 Report

The authors' responses and the changes that have been made are satisfatory. 

Author Response

Thank you very much for reviewing our paper.
We believe that the quality of our paper has improved thanks to your kind remarks.

In the abstract, there was no mention of women being included in the analysis.

We have corrected a part of the abstract as follows
Please check it.

Abstract(Lines 2)  The participants were 192 community-dwelling older adult women.
